# Synergistic Chromatin-Modifying Treatments Reactivate Latent HIV and Decrease Migration of Multiple Host-Cell Types

**DOI:** 10.3390/v13061097

**Published:** 2021-06-08

**Authors:** Alexandra Blanco, Tarun Mahajan, Robert A. Coronado, Kelly Ma, Dominic R. Demma, Roy D. Dar

**Affiliations:** 1Department of Bioengineering, University of Illinois at Urbana-Champaign, Urbana, IL 61801, USA; ablanco3@illinois.edu (A.B.); tarunm3@illinois.edu (T.M.); rac5@illinois.edu (R.A.C.); ycma2@illinois.edu (K.M.); dom10718@comcast.net (D.R.D.); 2Center for Biophysics and Quantitative Biology, University of Illinois at Urbana-Champaign, Urbana, IL 61801, USA; 3Carl R. Woese Institute for Genomic Biology, University of Illinois at Urbana-Champaign, Urbana, IL 61801, USA

**Keywords:** human immunodeficiency virus, latency reversal, monocytes, T cells, chromatin modifier, LTR promoter

## Abstract

Upon infection of its host cell, human immunodeficiency virus (HIV) establishes a quiescent and non-productive state capable of spontaneous reactivation. Diverse cell types harboring the provirus form a latent reservoir, constituting a major obstacle to curing HIV. Here, we investigate the effects of latency reversal agents (LRAs) in an HIV-infected THP-1 monocyte cell line in vitro. We demonstrate that leading drug treatments synergize activation of the HIV long terminal repeat (LTR) promoter. We establish a latency model in THP-1 monocytes using a replication incompetent HIV reporter vector with functional Tat, and show that chromatin modifiers synergize with a potent transcriptional activator to enhance HIV reactivation, similar to T-cells. Furthermore, leading reactivation cocktails are shown to differentially affect latency reactivation and surface expression of chemokine receptor type 4 (CXCR4), leading to altered host cell migration. This study investigates the effect of chromatin-modifying LRA treatments on HIV latent reactivation and cell migration in monocytes. As previously reported in T-cells, epigenetic mechanisms in monocytes contribute to controlling the relationship between latent reactivation and cell migration. Ultimately, advanced “Shock and Kill” therapy needs to successfully target and account for all host cell types represented in a complex and composite latency milieu.

## 1. Introduction

Human immunodeficiency virus (HIV) infects its host cell genome by stable integration of its DNA into either a highly active and expressive site where it actively replicates to produce viral progeny, or into an inactive or quiescent site where it is capable of spontaneously reactivating later (Figure 1A). Since only active replicating virus is targeted by antiretroviral therapy (ART), cells harboring latent provirus persist and form a stable and long-lived viral reservoir [1]. The reservoir of latently infected cells constitutes the major barrier to curing HIV [2]. Current strategies to eradicate the latent reservoir include “Shock and Kill”, in which the infected cells are treated with potent activating cocktails, termed latency reversal agents (LRAs), and purged into a productively replicating state where they can go through immune clearance [3,4]. Successful application of LRA cocktails would enable the removal of ART after eradication of the latent reservoir. Alternatively, the “Block and Lock” treatment strategy aims to force integrated provirus into a deep latent state where it can no longer reactivate spontaneously [5,6,7], and offers a functional cure where antiretroviral therapy can be removed from infected individuals [5,8]. Both therapeutic strategies are being actively pursued and are currently an open paradigm in researching a cure for HIV.

HIV latency is related to its gene expression [11,12]. Fluctuations in gene expression or “noise” in the activation of HIV play an important role in viral decision-making [11,12]. This suggests that compounds which modulate noise can be used in LRA cocktails to reactivate latent HIV [9]. Noise is caused by the random timing of molecular interactions during transcription and translation. Noise can be used to quantify episodic transcription of the HIV promoter from different integration sites across the genome [13,14]. Here both the HIV long terminal repeat (LTR) and housekeeping promoters transcribe in a highly discontinuous or bursty manner. Transcriptional bursting is characterized by burst size (the number of mRNA produced per activity pulse of an episodic promoter) and burst frequency (the initiation rate of the episodic promoter from an off state to an actively transcribing state) (Figure 1A–1B). As HIV is a bursty stochastic viral switch, noise-modulating compounds can synergize or suppress reactivation of latent HIV [9]. Noise enhancers are compounds that increase noise without affecting mean expression. They do this by increasing burst size while decreasing burst frequency. Consequently, noise enhancers synergize with transcriptional activators, which primarily modulate mean expression. Such synergy can offer large increments in gene expression (Figure 1C). The use of LRA drug cocktails consisting of noise enhancers combined with activator treatments can synergistically reactivate latent HIV and is proposed to complement existing “shock and kill” eradication strategies (Figure 1C) [9]. 

HIV infects multiple cell types including T-cells, monocytes, and macrophages. This complicates eradication strategies as the latent reservoir comprises these multiple cell types [15,16,17,18]. Here, regulation and dynamics of viral gene expression may vary on a cell type basis [19]. Therefore, it is critical to understand if HIV gene expression and latent reactivation are conserved between cell types. Specifically, it is of interest whether leading latency reactivation treatments are cell type specific or affect all cell types equally.

Here we establish monocytic cell lines harboring either an integrated HIV LTR promoter or full-length HIV to investigate the conservation of viral–host gene expression relationships between human T-cells and monocytes. Using a panel of treatments, we observe that the HIV LTR promoter consistently activates in both monocytes and T-cells when combining chromatin modifiers with a transcriptional activator. Furthermore, the same treatment combinations synergize reactivation of latency in both monocytes and Jurkat T-cells. Finally, consistent with previous studies in T-cells, we show that chromatin-modifying drug cocktails differentially control latent reactivation and cell migration in monocytes. These findings may have profound implications for advanced “Shock and Kill” therapies which need to successfully target and account for all cell types represented in mixed latent cell populations.

## 2. Materials and Methods

### 2.1. Cell Culture

Jurkats (naïve, JLat 9.2, Ld_2_G iso 41, iso 20, and iso 70) were cultured in RPMI 1640 media (Corning Inc., Corning, NY, USA) with L-glutamine and phenol red, supplemented with 10% fetal bovine serum (FBS), and 1% penicillin/streptomycin (P/S). THP-1 were grown in RPMI 1640 media (ATCC, Manassas, VA, USA) supplemented with 10% FBS, 1% P/S, and 0.05 mM 2-mercaptoethanol. HEK293T cells were cultured in DMEM media with 10% FBS and 1% P/S. Cells were incubated with 5% CO_2_ at 37 °C. 

### 2.2. Generation of Ld_2_G THP-1 Library

The LTR-d_2_GFP vector was packaged into HEK293 cells along with VSV-G and ΔR8.9 plasmids for lentivirus generation. HEK293 cells (60–70% confluent) were transfected using FuGene 6 transfection reagent (Promega, Madison, WI, USA) according to the manufacturer’s instructions. Viral supernatant was collected after 24 h and 48 h, centrifuged at 500× *g* to remove all remaining cells, and filtered through a 0.45 μm PES filter. THP-1 cells were infected with lentiviral supernatant at a multiplicity of infection (MOI) of 1 (i.e., MOI = 1) to minimize multiple integration sites. Viral supernatant and cells were centrifuged at 1200 rpm for 2 h at room temperature. After spinoculation, GFP+ cells were single-cell sorted into 96-well plates using a BD FACSAria II instrument. Cells were allowed to expand for over 3 weeks into clonal populations, creating a final library of 35 distinct LTR-d_2_GFP clones. After expansion, green fluorescence protein (GFP) expression of individual clones was measured by flow cytometry. Clones separated into two fluorescence intensity groups defined as low and high (Appendix A).

### 2.3. Establishment of a Monocytic Latency Model (TLat)

A THP-1 latency model was generated using the full-length HIV retroviral vector NL4-3 ΔEnv EGFP (HIV Reagent Program, Manassas, VA, USA). Lentivirus was generated by transfection of HEK293 cells using FuGene 6 transfection reagent according to the manufacturer’s instructions. Unconcentrated viral supernatant was used to infect 0.5 × 10^6^ cells/mL in a 6-well plate using 10 µg/mL polybrene (Sigma Aldrich, St. Louis, MO, USA) at a MOI = 1. Cells were allowed to grow and recover for ~1 week following infection. GFP cells were sorted by FACS and further cultured for ~1 week. This population was enriched for latently infected cells. Next, cells were stimulated with tumor necrosis factor alpha (TNF) for 24 h, and GFP+ single-cells were sorted into 96-well plates. Cells were allowed to expand for over 3 weeks into clonal populations. To measure reactivation percentage of each clonal population, cells were stimulated with TNF for 24 h. After TNF treatment, the percentage of GFP positive cells determined the reactivation percentage (Appendix A). Out of 14 clonal populations that grew out in the 96-well plates, only one clone, TLat 2C8, had basal reactivation of ~0.7% and resulted in reactivation of ~4% after TNF stimulation while the rest of the isoclones remained unresponsive to treatment. We termed the responsive clonal population the “TLat”. 

### 2.4. Detection of Human Immunodeficiency Virus (HIV) DNA by Polymerase Chain Reaction (PCR) and Agarose Gel Electrophoresis

Total DNA from TLat 2C8 (4 × 10^6^ cells) was extracted with the DNeasy Blood and Tissue Kit (Qiagen, Germantown, MD, USA). DNA purity and concentration were measured with a Nanodrop One instrument (ThermoFisher Scientific, Waltham, MA, USA). Polymerase chain reaction (PCR) was performed using the Phire Hot Start II PCR Master Mix (ThermoFisher Scientific, Waltham, MA, USA) according to the manufacturer’s instructions. Briefly, 16 ng template DNA was subjected to 35 cycles of PCR in a S100 Thermal Cycler (Bio-rad. Hercules, CA, USA) in a total volume of 20 µL containing the master mix (with Phire Hot Start II DNA polymerase, 1.5 mM MgCl_2_ and 200 µM of each dNTP), 0.5 µM of each primer, and 2.5% DMSO. Each cycle comprised a 30 s denaturing step (98 °C), a 5 s annealing step (64.9 °C), and a 10 s extension (72 °C). The following primers were used to amplify a ~2.9 kbp region of the HIV *gag* gene: forward primer (5’-AGAGCGTCGGTATTAAGCGG-3’), and reverse primer (5’-CTGTCCACCATGCTTCCCAT-3’).

Agarose gel electrophoresis was performed to visualize the amplified fragments. Briefly, a 1% agarose gel in 1× tris-acetate-EDTA (TAE) buffer (ThermoFisher Scientific, Waltham, MA, USA) was poured in an Owl EasyCast B2 gel electrophoresis instrument (ThermoFisher Scientific, Waltham, MA, USA) and allowed to solidify for 30 min. Samples were loaded onto each well with 6X purple dye (New England Biolabs, Ipswich, MA, USA). Lambda DNA-HindIII Digest and 100 bp ladder (New England Biolabs, Ipswich, MA, USA) were used as DNA markers. Gel electrophoresis was run at 60 V for 90 min in 1× TAE buffer. After electrophoresis, the gel was stained with SYBR Green (ThermoFisher Scientific, Waltham, MA, USA) for 30 min and amplified fragments of the correct size were imaged using blue light illumination with an Axygen Gd Gel Documentation System (Corning Inc., Corning, NY, USA) (Appendix A). 

### 2.5. Drug Treatments

Over-expression of Ld_2_G cell lines and latency reactivation assays were performed with TNF (R&D Systems, Minneapolis, MN, USA) at a final concentration of 10 ng/mL. All chromatin-modifying agents were acquired from Cayman Chemicals (Ann Arbor, MI, USA). Trichostatin A (TSA) was used at a final concentration of 400 nM, JQ1 at 1 µM, suberoylanilide hydroxamic acid (SAHA, or vorinostat) at 2.5 µM, 5-Aza-2-deoxycytidine (5-Aza) at 5 µM, and valproic acid (VPA) at 1 mM. For migration experiments, JLat 9.2 was treated with phorbol 12-myristate 13-acetate (PMA), a protein kinase C agonist, at 200 ng/mL. 

### 2.6. CXCR4 Antibody Staining

For detection of chemokine receptor type 4 (CXCR4) surface expression, cells were stained with a phycoerythrin (PE)-conjugated CD184 (CXCR4) monoclonal antibody (ThermoFisher Scientific, Waltham, MA, USA) according to the manufacturer’s instructions. Briefly, 5 µL of pre-titrated antibody was added to cell samples at a final volume of 100 µL. The staining was performed on ice for 30 min in the dark. Cells were washed at least once to remove unbound antibody and analyzed using flow cytometry.

### 2.7. Migration Assay

Migration assays shown in Appendix A were performed in 24-well transwell plates consisting of 5 μm pore polycarbonate membranes (Corning Inc., Corning, NY, USA). JLat 9.2 or TLat cells were grown to a density of ~1 × 10^6^ cells/mL and 1 mL of cell suspension was transferred to a 24-well plate and stimulated with single or combination treatments for 24 h. After incubation, cells were pelleted by centrifugation (500× *g*) and resuspended in migration medium (RPMI 1640 with L-glutamine and 25 mM HEPES supplemented with 0.5% bovine serum albumin (Sigma-Aldrich, St. Louis, MO, USA)) at a concentration of 1.5 × 10^6^ cells/mL. 0.6 mL of migration medium was added to the lower chamber of the transwell, followed by addition of SDF-1α (R&D Systems, Minneapolis, MN, USA) at a concentration of 25 ng/mL. Transwells were loaded with 200 µL of diluted cells and inserted on top of the lower chamber. Air bubbles under each transwell were carefully removed. Cells were allowed to migrate for 3 h at 37 °C in 5% CO_2_. Migrating cells (cells in the lower well) were counted using a MOXI Z cell counter (ORFLO, Ketchum, ID, USA) and analyzed for reactivation via flow cytometry. Results of the migration assays are shown in Appendix A.

### 2.8. Flow Cytometry Analyses and Gating Strategy

GFP fluorescence, reactivation percentage, and CXCR4 expression were quantified using a BD LSR Fortessa II analyzer. Based on forward and side scatter, gates were set up to exclude debris and filter out single events that consist of two independent particles. 30,000 cells in total were collected. All flow cytometry data were analyzed using FCS Express 6 (Appendix A). A gate for JLat or TLat reactivation in the side scatter vs. fluorescein isothiocyanate (FITC) scatter plot was created using the untreated measurement similar to previous reports [6,20]. 

### 2.9. Gene Expression Noise Calculations

For noise analyses of the Ld_2_G Jurkat and THP-1 cell lines, FCS Express 6 was used to analyze a region of interest (ROI) containing the highest concentration of cells from each treatment sample. This ROI contains ~6% cells (1800 or 3000) of total live cells (30,000 or 50,000) with similar cell size in the forward scatter, and it has been used to filter out extrinsic noise [21]. The mean and the variance of the fluorescence within the ROI are calculated from each treatment. Values are corrected using the following equations:*μ*_corr_ = *μ*_treat_ − *μ*_N_
*σ*_corr_^2^ = *σ*_treat_^2^ + *σ*_N_^2^
where *μ*_corr_ and *σ*_corr_^2^ stand for the corrected mean and variance for each treatment. *μ*_treat_ and *σ*_treat_^2^ stand for the mean and variance for each treatment, and *μ*_N_ and *σ*_N_^2^ stand for the mean and variance of a non-fluorescent sample (naïve Jurkat or THP-1). The sample mean fluorescence is *μ*_corr_ and the sample CV^2^, or noise, is calculated as:*σ*_corr_^2^ = *σ*_treat_^2^ + *σ*_N_^2^.

### 2.10. Bliss Independence Score Calculation

Excess over Bliss (EoB) scores were calculated for each treatment to determine whether a drug’s interaction with TNF is additive or non-additive [6,9,22]. The EoB is defined as the difference between the observed combined effect and the expected combined effect:*EoB*_d_ = *f*_TNF+d_ − *E*[*f*_TNF+d_] = *f*_TNF+d_ − (*f*_TNF_ + *f*_d_ − *f*_TNF_*f*_d_)
where *f*_TNF_ and *f*_d_ denote the reactivation percentage (or GFP fluorescence) of the population when adding TNF and drug by themselves, and *f*_TNF+d_ denote the reactivation (or GFP fluorescence) when the drug and TNF are added together. Positive EoB values indicate synergy (represented with “S” in bar plots). 

### 2.11. Two-State Model Simulations 

We use the two-state model of transcriptional bursting (Figure 1B). This model has been previously validated for HIV gene expression [13]. This is a discrete Markov jump process, where the promoter switches between two discrete states—high and low transcriptional activity. Without loss of generality, we assume that in the low state there is no transcriptional activity. The model is dictated by six kinetic parameters: (1) rate of initiation into the transcriptionally active state, *k*_on_; (2) rate of switching back into the transcriptionally inactive state from the active state, *k*_off_; (3) rate of transcription from the active state, *k*_m_; (4) degradation rate for mRNA (*M*), *γ*_m_; (5) rate of translation, *k*_p_; and (6) degradation rate for protein (*P*),*γ*_p_.

Broadly, we consider two simulation experiments: (1) single-cell time-series trajectories; and (2) noise-vs-mean plots. For the first experiment, we compare single-cell time-series trajectories between four scenarios—untreated (UN), noise enhancer (NE), activator (AC) and AC plus NE (AC + NE). The kinetic parameters used for the four scenarios are given in Appendix A. NE is obtained by increasing burst size, while keeping mean expression constant compared to UN. This is achieved by decreasing both *k*_on_ and *k*_off_. To get an AC, we increase burst frequency by increasing *k*_on_. For NE + AC, we increase both burst size and frequency by decreasing *k*_off_ but increasing *k*_on_. We set an arbitrary threshold of 300,000 proteins for active replication; the threshold is defined for illustrative purposes and was taken from Dar et al. [9]. These single-cell time-series trajectories were used in Figure 1C.

For the noise-vs.-mean plots (Figure 1D and Appendix A), we modulate burst size and frequency together on a grid (Appendix A). For each combination over the 2D grid, we simulate the two-state model and compute noise, mean expression and the percentage of cells which cross the active replication threshold. We consider two cases: (1) modulating burst size by changing *k*_m_ (Figure 1D); (2) modulating burst size by changing *k*_off_ (Appendix A).

Since the goal of the study is to identify reactivation synergy, we give an expression-based definition of synergy. Assume that *<P>* is the mean abundance for protein molecules. Then, for synergy, the following relationship must hold:⟨*P*⟩_UN_ = ⟨*P*⟩_NE_ < ⟨*P*⟩_AC_ < ⟨*P*⟩_AC+NE_
This definition was adapted from Dar et al. [9]

The two-state model was simulated exactly using Gillespie’s stochastic simulation algorithm (SSA) [23]. We use the R package GillespieSSA2 for all the simulations [24]. The code used for these simulations is provided in a Jupyter notebook (https://github.com/Tarun-Mahajan/Stochastic_Simulation_HIV_Noise).

### 2.12. Statistical Methods

We assume the underlying population follows a Gaussian distribution. The mean of the population (µ) equals the mean of the sampling distribution (µ*_x_*), and the standard error of the sampling mean (σ*_x_*) can be approximated by:σ*_x_* = σ/ sqrt(n)
where σ is the standard deviation of the population, and n represents the sample size. The mean is represented as an average of two or more separate measurements and the standard error is represented as error bars. 

## 3. Results

### 3.1. Stochastic Simulations Reveal an Over-Expression Strategy with Noise Enhancement

Previous investigations have shown that the HIV LTR promoter displays episodic transcriptional bursting and that its decision making between active replication and latency is governed by stochastic fluctuations in viral gene expression [11,12,13,14] (Figure 1). In addition it has been shown that noise modulating compounds, including chromatin modifiers, are capable of synergizing with activators to generate more activation [6,9]. We simulated the two-state model of transcriptional bursting consisting of a promoter that can switch between a transcribing and non-transcribing state (Figure 1B) [14,25,26]. Simulations show that the two-state model allows synergy in gene expression of a gene product (e.g., such as Tat). Since reactivation is related to such gene expression, this can lead to synergy in latent reactivation. Treatment with an activator (blue, Figure 1C) alone causes only a few cells to cross the active replication threshold (protein count at which Tat positive feedback is initiated and latent HIV is reactivated). However, the combination (red, Figure 1C) of an activator and noise enhancer (purple, Figure 1C) exhibits synergy causing a large proportion of the cells to cross the threshold [9]. Here, the noise enhancer increases transcriptional burst size (k_m_/k_off_), while keeping mean expression constant. 

We also performed extensive simulations to explore the dependence of reactivation (% cells crossing active replication threshold) on changes in burst size and frequency in the noise-vs.-mean space (Figure 1D). Burst size and frequency were modulated over a 2D grid (Methods). Changes in burst frequency move along the iso-burst lines (dashed) in the noise-vs-mean space. % cells that surpass the threshold increases with both burst size and frequency increases (colorbar, Figure 1D).

### 3.2. Generation of Single-Integration Clones to Study HIV Promoter Expression in Monocytes 

To quantify the expression of the HIV LTR promoter in THP-1 monocytes, we generated a LTR-d_2_GFP (Ld_2_G) THP-1 library based on previously established Jurkat T cell lines [13,14]. THP-1 cells were infected with a vector consisting of the HIV-1 LTR promoter expressing a short-lived green fluorescent protein (GFP) reporter (d_2_GFP) with a 2.5 hour half-life (Figure 1E) [13,14,27]. Cells were infected at a MOI = 1 to ensure a single integration in each cell. GFP+ cells were individually sorted by FACS into 96-well plates and grown to generate a library resulting in 35 isoclonal populations. The GFP expression distribution for each cell line was determined by flow cytometry. The library displayed a diverse range of expression mean, encompassing two orders of magnitude (Appendix A). Based on this characterization, we then selected two separate clones (a low or “E7” clone, and a high or “E6” clone) for additional measurements. 

We explored the effects of TNF, an activator, and chromatin modifying agents on the two separate clones. Both untreated clones displayed a noise versus mean dependence consistent with a constant burst size model line as previously reported [14,28] (Figure 1F). Chromatin-modifying treatments increased both noise and mean, which is consistent with a burst size increase and a decrease in burst frequency. TNF decreased noise while increasing mean, which is consistent with an increase in both burst size and burst frequency [13,28,29,30]. The results suggest that in THP-1 monocytes, similar to Jurkat T-cells (Appendix A) [9], chromatin modifiers enhance noise and can potentially synergize over expression of the LTR and reactivation of HIV from latency when combined with activators of transcription.

### 3.3. Chromatin Modifiers Combined with Activators Enhance HIV Promoter Expression in Both Monocytes and T-Cells

To test whether chromatin-modifying treatments combined with activators would enhance HIV promoter expression in monocytes, we subjected two Ld_2_G THP-1 cell lines (clones “E7” and “E6”) to a variety of drug treatments involving chromatin remodeling. These treatments include histone deacetylase (HDAC) inhibitors (TSA, SAHA, and VPA), a methylation inhibitor (5-Aza), and a bromodomain and extraterminal domain (BET) protein inhibitor (JQ1), and their combination with the transcriptional activator TNF. Drug treatments were carried out for 24 h using previously published concentrations [9]. JQ1 strongly synergized with TNF to enhance HIV expression in Ld_2_G monocytes (Figure 2A) consistent with previous reports [10]. JQ1 is highly specific for the BET-containing protein BRD4, which binds to the positive-transcription elongation factor b (P-TEFb) in the absence of Tat [31]. Furthermore, JQ1 has been demonstrated to act as a positive regulator of HIV reactivation from latency [10,32]. TSA, VPA, and SAHA promote HIV transcriptional activation, HIV gene expression, and latency reversal [33,34]. Subsequently, these have been reported to synergistically reactivate latency when combined with TNF [35]. We show that these HDAC inhibitors synergized with TNF to enhance HIV promoter expression in Ld_2_G monocytes (Figure 2). Additionally, 5-Aza also synergized with TNF to enhance HIV promoter expression (Figure 2A, left). 5-Aza facilitates HIV transcription by challenging hypermethylation of the HIV promoter [36,37]. 

To confirm HIV promoter activation in other cell types, we used previously established Jurkat T cell lines harboring the LTR-d_2_GFP construct [13,14]. We subjected three cell lines of varying intensity, each harboring a unique LTR genomic integration site, to the same treatments described above. Similar to monocytes, combination treatments between TNF and chromatin-modifying agents synergized to activate the HIV promoter across T-cell lines (Figure 2B). Here, the synergy observed for a noise-enhancing chromatin remodeling treatment combined with a transcriptional activator is consistent with the simulations for episodic transcription and noise modulating cocktail strategy (Figure 1) [9]. The synergistic effect produced by combining a transcriptional activator with a chromatin modifying compound implies that drug treatments can cause simultaneous activation of the HIV LTR promoter in more than one cell type.

### 3.4. Establishment of an In Vitro Model of HIV-1 Latency in Monocytes (TLat)

To test if our selection of chromatin modifiers synergize with transcriptional activators to enhance latent reactivation, we generated a latency model in THP-1 cells based on the well-studied in vitro latency model in Jurkat T cells (JLat) (Figure 3A) [38] (Methods). We used a construct containing a full-length HIV-1 vector with a non-functional Env gene and a reporter gene for enhanced green fluorescent protein (EGFP) (NL 4-3 ΔEnv EGFP). We infected a culture of THP-1 monocytes with lentivirus containing this vector at a MOI = 1. Next, we expanded the infected population and isolated GFP negative cells by FACS. This isolated population contains a mixture of non-infected cells and latently infected cells. To enrich for latently infected cells, we treated this population with TNF for 24 h to activate HIV transcription. Following treatment, GFP positive cells were isolated into single cells in four 96-well plates by FACS and allowed to expand for 3–4 weeks to generate clonal populations. A clonal THP-1 latency cell line (TLat 2C8) was used to further characterize reactivation from latency.

### 3.5. Synergistic Cocktails Enhance Reactivation from Latency

The synergistic effect seen in Ld_2_G cell lines indicates that these treatments could synergize to enhance reactivation of latent HIV (Figure 1 and Figure 2). To test if chromatin modifiers reactivate latency in monocytes, we treated the TLat with a series of drug cocktails. We used four previously described chromatin modifiers, including HDAC and BET protein inhibitors, and tested their potential as latency reversal agents (LRAs) alone and in combination with TNF. These cocktails have been shown to enhance latency reactivation in Jurkat T cells [9,10,35,39]. The chromatin modifiers alone reactivated latency in the range of 0.8–3.9%. All combinations between chromatin modifiers and TNF resulted in synergistic reactivation, in the range of 6.4–15.4% (Figure 3B, left). Earlier, we reported that JQ1 and TNF synergized to enhance HIV promoter expression across two different integration sites in monocytes. Consistent with this result, JQ1 and TNF synergized to provide the highest reactivation from latency (15.4%) compared to the other drug treatments (Appendix A). This is nearly a 4-fold increase in latency reactivation when compared to TNF alone. SAHA and VPA (HDAC inhibitors) synergized with TNF and provided the second to highest reactivation percentage. 

We validated our results using the well-studied in vitro Jurkat T-cell latency model [9,37,38] (JLat 9.2) (Figure 3B, right). We measured the synergistic effects of TNF and chromatin modifiers on JLat reactivation percentage after 24 h treatments. While TNF reactivates ~15% of cells, chromatin-modifying agents alone provide very little to no reactivation (~0.2, on average). All combination treatments between chromatin modifiers and TNF synergized to enhance T-cell reactivation in the range of 30.3–58.6%. The HDAC inhibitors SAHA and VPA synergized with TNF to provide the highest reactivation from latency (50.5% and 58.6%, respectively). We showed that these combinations also resulted in high synergistic reactivation from latency in the TLat (Figure 3B, left).

Further, our data exhibited a positive correlation between GFP fluorescence and reactivation percentage in both monocytes and T-cells (Figure 3C). This indicates a monotonic relationship between the amount of reactivation and mean GFP expression of the reactivated cells. Synergistic combinations between chromatin modifiers and TNF yield higher mean expression and reactivation percentages than each of these alone. 

### 3.6. Leading Drug Cocktails Affect Host Cell CXCR4 Surface Expression

Understanding viral–host interactions and regulatory mechanisms is crucial for the development of successful clinical therapies. One viral-host regulatory interaction that could challenge “Shock and Kill” is the co-expression between C-X-C chemokine receptor type 4 (CXCR4) and the HIV LTR promoter [20]. Stromal cell-derived factor 1α (SDF-1α) signaling via CXCR4 regulates a variety of cellular functions, including T-cell chemotaxis, adhesion, lymphocyte development, homeostasis, and cell-cycle progression [40,41]. CXCR4 also functions as a co-receptor for HIV-1 entry [42]. CXCR4-LTR co-expression presents a risk for latency reversal as certain LRAs could induce cell migration to SDF-1α-rich areas by over-expression of CXCR4, where uninfected cells could potentially become the target of new active and latent infections [20,43,44,45]. Here, cell migration is affected by both the direct influence of LRA treatment on surface CXCR4 as well as co-expression of the CXCR4 and the HIV LTR promoters.

To explore the effects that leading reactivation cocktails have on monocyte cell migration, latently infected cells (TLat) were subjected to previously described reactivation cocktails for 24 h. CXCR4 surface expression was evaluated using a conjugated monoclonal antibody. Reactivation percentage as well as CXCR4 surface expression were quantified via flow cytometry. Based on relative changes in CXCR4 from the untreated control, cell populations fall into three distinct classes of drug cocktails (Figure 4A): (1) reactivation cocktails (purple triangles), (2) migration-modifying cocktails (red diamonds), and (3) reactivation and migration-modifying cocktails (blue circles). Migration-modifying cocktails change CXCR4 surface expression while offering low levels of reactivation. All chromatin modifying agents fell into this category. Reactivation cocktails provide latency reversal without changing CXCR4 surface expression from the untreated cells. Both TNF and TNF + TSA fall into this category. Reactivation and migration modifying cocktails decrease CXCR4 surface expression while offering high reactivation levels. The synergistic combination cocktails between SAHA, VPA, and JQ1 with TNF fell into this category. Cells treated with cocktails in this class are shown to have decreased CXCR4 levels from the untreated control. These observations suggest that increased levels of reactivation cause surface CXCR4 expression to decrease. Bohn-Wippert et al. similarly reported this phenomenon in JLat cell lines, where leading drug cocktails were shown to differentially control migration and reactivation of latently infected T-cells [20]. 

To explore our findings in a different cell type, we subjected latently infected T-cells (JLat 9.2) to leading reactivation cocktails and measured CXCR4 surface expression after treatment (Figure 4B). Similarly, we report that cell populations fall into the same three categories described earlier with similar trends as in the monocytes. Chromatin modifiers alone act as migration modifying cocktails while TNF falls into the reactivation category. All synergistic drug combinations act as reactivation and migration modifying cocktails. We observe a similar trend in reduction of CXCR4 surface expression with increasing reactivation for a majority of the synergistic cocktails. We also show that synergistic LRA cocktails decrease SDF-1α mediated cell migration across monocytes and T-cells (Appendix A). Furthermore, as a control, we test CXCR4 levels of treated uninfected naïve cells to show that CXCR4 levels are altered by viral reactivation in addition to the direct influence of LRA treatment (Appendix A).

## 4. Discussion

Anatomical reservoirs of HIV comprise multiple cell types. Transcriptionally silent provirus in these latently infected cells may spontaneously reactivate to start producing infectious virions [46]. While resting memory CD4 T-cells are the best-characterized latent reservoir [19], the mechanisms of latency in relevant immune cell types such as monocytes are still poorly understood [15]. Circulating HIV-infected monocytes pose a high risk to viral dissemination. Upon activation, monocytes enter tissues and differentiate into macrophages, seeding additional tissue reservoirs by targeting uninfected cells [47]. Being responsible for a variety of immune functions and given their role in HIV pathogenesis, further understanding the mechanisms of latency within these cells is crucial for the development and guidance of clinical therapies [48]. We highlight the importance of chromatin remodeling and epigenetics in the context of HIV gene expression, latent reactivation, and “Shock and Kill” therapy. 

We identified compounds that synergize with TNF to over-express HIV by monitoring LTR promoter activity in monocytes and T-cells. In both cell types, chromatin modifiers mostly displayed gene expression noise enhancement with an increase of transcriptional burst size while TNF activation increased burst size and burst frequency [9,28] (Figure 1). This noise enhancement provides a signature to identify synergistic drug treatments for HIV [9]. These compounds include LRAs that affect chromatin configuration by inhibiting histone deacetylation, bromodomain proteins, and DNA methylation. This is particularly useful in the case of latency reversal, as several of these mechanisms contribute to post-integration latency and silencing of HIV gene expression [49]. Combinations of transcriptional activators and chromatin modifiers leading to synergistic over-expression activate transcription through TNF-induced NF-kB activation and permissive epigenetic modifications [35,50,51] (Figure 2). Adam et al., explained that this synergy is caused by the ability of HDAC inhibitors to prolong TNF-induced NF-kB DNA-binding activity and NF-kB nuclear localization in HeLa cells [50]. Our findings, together with these previous studies, demonstrate the importance of chromatin context in regulating HIV gene expression in two relevant cell types.

Synergistic activation of the LTR promoter in monocytes has important implications for “Shock and Kill”. LRA cocktails can either target a specific cell type or globally affect all cell types in the latent reservoir. However, it has been shown that LRA cocktails do not globally affect all latently infected cells equally [52]. To evaluate the potential that the LRAs have on latent reactivation in monocytes, we established an in vitro monocytic model of latency (TLat). We used an HIV retroviral vector to create a latently-infected reporter cell line similar to the well-established in vitro Jurkat T-cell latency model (JLat) [38]. Our model response to TNF stimulation is consistent with a previously reported THP-1 cell line that engineered and integrated replication-competent p89.6 EGFP [53]. 

Commonly used models to study HIV latency in cells of the mononuclear phagocyte lineage originate from chronically infected cell lines [54]. Among these, the U1 promonocytic cell line has been defined as a model of *Tat*-dependent HIV latency. This cell line was generated from the surviving population of a chronically infected U937 promonocyte clone [55] and it has been used as a model for studying post-integration latency [56]. It produces low levels of HIV at baseline, and its expression can be dramatically upregulated by treatment with cytokines [57,58]. This promonocyte cell line contains a mutation in the *Tat* protein and its relative state of proviral latency has been linked to a defective Tat/TAR interaction [59], which limits its potential as a useful in vitro latency model. Similar to the TLat, latent reactivation of this cell line has been achieved by JQ1 and synergistic combinations of HDAC inhibitors and TNF [10,51,60]. Despite its usefulness for transcriptional studies, the *Tat* protein mutation in the U1 cell line limits its relevance to study latency [60]. Therefore, since it contains a latently integrated full-length HIV genome (Δ*Env*) with a functional *Tat*-TAR axis, the TLat cell line reported here may serve as a more promising model for studying latency in monocytes along with other existing models [53].

We confirmed synergistic latent reactivation from chromatin modifier and activator combinations using the TLat cell line, where JQ1, SAHA, and VPA provided the strongest synergy with TNF to reactivate latent HIV (Figure 3). The synergistic cocktails employed here are similar to combinations that have been tested in a variety of in vitro cell line models of latency. They have been shown to induce HIV gene expression from patient-derived latently infected cells [39,61]. The strong synergistic effect obtained from TNF and JQ1 combinations in both latently infected and minimal HIV promoter cell lines suggests the importance of bromodomain proteins, especially BRD4, in monocyte regulation of HIV gene expression. Binding of *Tat* to P-TEFb at the HIV promoter is necessary for transactivation of HIV transcription [62]. The P-TEFb heterodimer complex is made up of two host proteins, cyclin T1 (CycT1) and cyclin-dependent kinase 9 (CDK9). Since BRD4 has high specificity for both subunits of P-TEFb, it competes with *Tat* for binding of P-TEFb at the HIV promoter [63]. Furthermore, low levels of CycT1 and phosphorylated CDK9 in monocytes render P-TEFb nonfunctional [64,65,66] contributing to restricted HIV gene expression. As JQ1 inhibits BRD4, *Tat* no longer competes for P-TEFb binding, allowing *Tat*-activated HIV transcription [31]. Furthermore, the strong synergistic effect seen between HDAC inhibitors and TNF in the reactivation of latent HIV confirms the relevance of HDAC-mediated regulation of HIV gene expression in monocytes and T-cells. Chromatin organization at the LTR and nucleosome 1 occupancy downstream of the transcriptional start site are some of the mechanisms reported to regulate latency in cell lines and in vitro primary cell models [33,36,60,67]. Inhibition of HDAC-mediated chromatin condensation at the HIV promoter stimulates viral transcription of latently infected cells [68]. Collectively, we show that epigenetic modifications at the HIV promoter regulate latency reversal in two different cell types. While in vitro cell line models of latency do not accurately recapitulate the ex vivo behavior of latently infected cells, they are useful for testing and selecting drug candidates for HIV therapy [39]. For instance, a study performed by Darcis et al., used latently infected cell lines to show that combinations between PKC agonists and P-TEFb releasing agents are highly promising LRA candidates. These combinations were shown to potently induce HIV gene expression in ex vivo cultures from cART-treated HIV-1+ aviremic patients [61].

HIV gene expression is controlled by complex viral-host regulatory mechanisms that vary on a cell type basis [69]. One such regulatory mechanism is viral control of cell migration [20,70,71,72]. Consistent with previous findings in T-cells [20], we show that latency reversal treatments affect cellular migration by altering CXCR4 surface expression in monocytes (Figure 4). Cells with greater reactivation levels show decreased CXCR4 surface expression and cell migration (Figure 4 and Appendix A). The observation of decreased cell migration of reactivated cells agrees with in vivo findings reporting that HIV-infected cells have decreased velocity and increased arrest coefficient [72]. This is a mechanism by which HIV may enhance viral fitness, as decreased cell migration would facilitate cell-to-cell transmission through virological synapses [73]. This phenomenon can be partially attributed to CXCR4 downregulation by the viral trans-activator of transcription (*Tat*) protein [74,75] or the viral protein *nef* [76,77]. A population of cells with higher reactivation percentage will have increased *Tat* and *nef*, which can then downregulate or bind to surface CXCR4 and antagonize its function [75,78]. This emphasizes the importance of studying the conservation of viral-host interactions at the cell surface that can inhibit CXCR4, decrease cell motility, and potentially affect HIV pathogenesis. Additional experiments on other relevant migration axes in addition to CXCR4 would be of interest to investigate the influence of HIV reactivation on cell migration. Collectively, these findings show a robust conservation of transcriptional regulation for the HIV LTR promoter, full-length HIV, and regulation of CXCR4 surface expression in both monocytes and T-cells. Additional studies accounting for other relevant immune cell types such as macrophages and their polarized phenotypes are needed to identify cell type-specific regulation of HIV gene expression and their mechanisms of latency.

HIV latency is established and maintained through complex mechanisms in diverse cell types [79]. Further understanding viral–host mechanisms that regulate HIV gene expression in all of its cell contexts is necessary to fully eradicate or control the latent reservoir. We show that epigenetic modifications of the HIV LTR promoter lead to permissive viral gene expression in monocytes. By establishing a post-integration model of HIV latency in monocytes, we show that combinations of the transcriptional activator TNF and chromatin-modifying agents synergize to enhance latent reactivation. We highlight the importance of these latency reversal treatments in the context of “Shock and Kill” and their implications for cell migration. Similarly, viral–host regulatory mechanisms in different cell types may also be important for the “Block and Lock” treatment strategy aimed to silence transcription and suppress reactivation from latency. Additional experiments incorporating in vitro co-cultures of mixed latent cell populations will reveal whether shared treatments can target multiple cell types. Fully eradicating or controlling the latent reservoir will require a deeper understanding of the conservation of viral-host relationships and the applicability of drug treatments on multiple cell types.

## Figures and Tables

**Figure 1 viruses-13-01097-f001:**
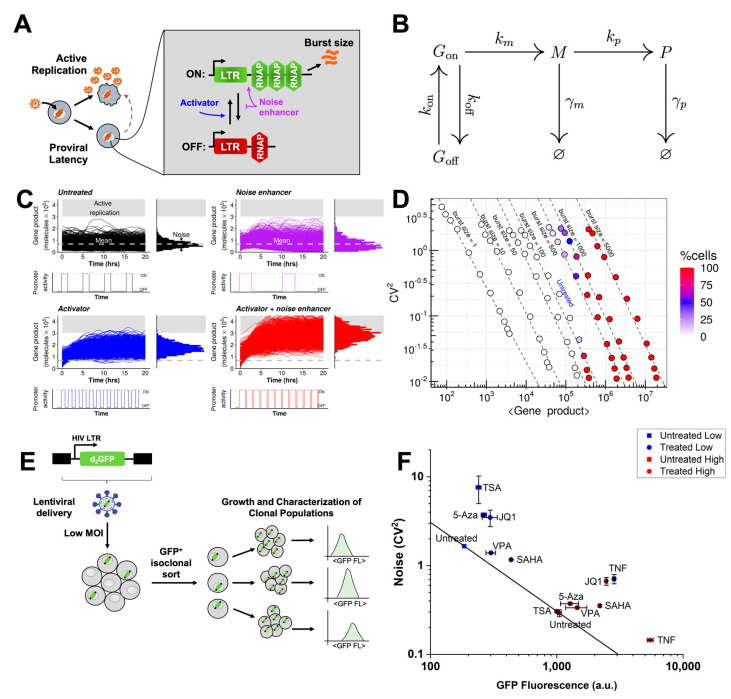
Model of episodic transcription predicts enhanced expression levels between compounds that increase transcriptional burst size and transcriptional burst frequency. (**A**) Schematic of human immunodeficiency virus (HIV) active replication, proviral latency, and latent reactivation (dashed arrow). (Inset) The HIV promoter transitions between an “off” state and a transcriptionally permissive “on” state at rates k_on_ (upward arrow) and k_off_ (downward arrow), respectively, and transcribes with rate k_m_ (horizontal arrow) in the “on state”. Activators of the long terminal repeat (LTR, blue) modulate k_on_ to increase expression rate, whereas adjusting k_m_/k_off_ (purple) primarily modulates expression noise. (**B**) The two-state model of transcriptional bursting for gene expression. The promoter is either in the on state (G_on_) or the off state (G_off_). Transcription of mRNA (M) only happens from the on state. Protein (P) is translated from the mRNA. Both mRNA and protein undergo degradation. The kinetic parameters for each transition reaction are also mentioned. The different parameters in the model are—k_on_: promoter on rate; k_off_: promoter off rate; k_m_: rate of transcription from the on state; k_p_: rate of translation; γ_m_: mRNA degradation rate and γ_p_: protein degradation rate. (**C**) Simulations exhibiting the synergistic effect of combining a noise enhancer with an activator. These simulations were performed for the two-state model of transcriptional bursting for the LTR promoter. Each sub-panel shows single-cell time series trajectories as line traces (left) and steady-state distribution for protein abundance (right) for four scenarios: untreated (black), noise enhancer (purple), activator (blue) and activator + noise enhancer (red). The arbitrary threshold for active replication of HIV is shown by the gray region. The untreated and noise enhancer samples show no cells cross the threshold. For the activator alone, only a few cells cross the threshold. However, the combination of the activator and the enhancer see a huge proportion of the cells crossing the threshold. Parameter values for each of the sub-panels are given in Appendix A. Each sub-panel shows 1000 different single-cell time trajectories (left). (**D**) Noise (CV^2^) vs. mean map for different combinations of burst size and frequency for the two-state model of transcriptional bursting for the LTR promoter. Each sphere in the plot represents steady-state mean and noise values calculated from 1000 different single-cells for a specific combination of burst size and frequency. Iso-burst lines are marked with dashed lines. For each iso-burst line, the same set of burst frequencies were simulated. As the burst size increases, the burst frequency at which reactivation starts decreases. For high enough burst size, as burst frequency increases, % of cells which cross the active replication threshold also increases. The set of burst sizes and frequencies which were used are given in Appendix A. Burst size was modulated by varying k_m_, while burst frequency was modulated by varying k_on_. (**E**) A copy of the HIV LTR promoter driving a short-lived green fluorescent protein reporter (d_2_GFP) was integrated into THP-1 monocytes. We refer to the newly-generated cell line as Ld_2_G THP-1. Ld_2_G THP-1 cells were individually sorted to generate a library in which each cell harbors a unique genomic integration site. The generated library comprises 35 isoclonal populations. The library was characterized based on LTR expression (GFP fluorescence) from each clone measured by flow cytometry. Based on the characterization, two separate clones were selected for subsequent measurements (E7 or “low” and E6 or “high”, Appendix A). (**F**) Noise-mean quantification for two THP-1 Ld_2_G isoclones under diverse chromatin modifying and activator treatments. In general, chromatin-modifying treatments increase noise (transcriptional burst size increase with burst frequency decrease) and mean while tumor necrosis factor alpha (TNF) decreases noise with an increase in mean (increase in transcriptional burst size and transcriptional burst frequency). These noise shifts in monocytes are consistent with previously reported noise shifts of LTR gene expression in T-cells [9,10]. Experiments were performed in duplicate and mean and standard error were plotted.

**Figure 2 viruses-13-01097-f002:**
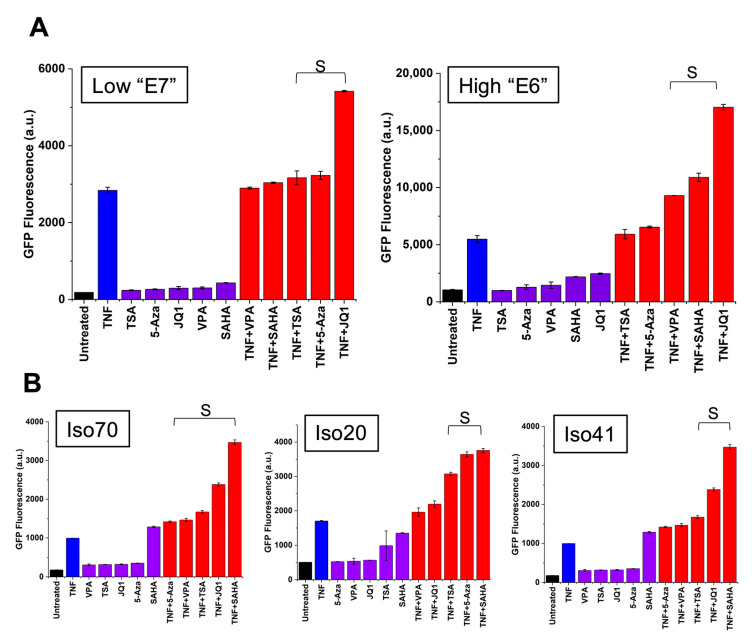
Chromatin modifiers synergize with TNF to activate the HIV promoter in monocytes. (**A**) To test if chromatin modifiers synergize with TNF to activate the HIV LTR promoter in monocytes, Ld_2_G THP-1 cells were exposed to TNF, four known chromatin modifiers (JQ1, suberoylanilide hydroxamic acid (SAHA), trichostatin A (TSA), and valproic acid (VPA)), and their combination treatments. The combination treatment between TNF and JQ1 shows the greatest synergistic activation of the LTR promoter in Ld2G monocytes across different genomic integrations, as evidenced by the non-additive increase in GFP fluorescence (<GFP FL>). Synergy was calculated from EoB. “S” denotes synergy. Experiments were performed in duplicate and mean and standard error were plotted. (**B**) To confirm LTR promoter activation across cell types, we subjected Ld_2_G Jurkat cell lines to treatments with TNF and chromatin modifiers. In a low expressing cell line, all combinations between TNF and chromatin modifiers synergize to enhance HIV expression (Left). Three chromatin modifiers (TSA, JQ1, and SAHA) similarly synergized with TNF to enhance LTR activation in a high expressing clone (Right). Synergy was calculated from EoB. “S” denotes synergy. Experiments were performed in duplicate and mean and standard error were plotted.

**Figure 3 viruses-13-01097-f003:**
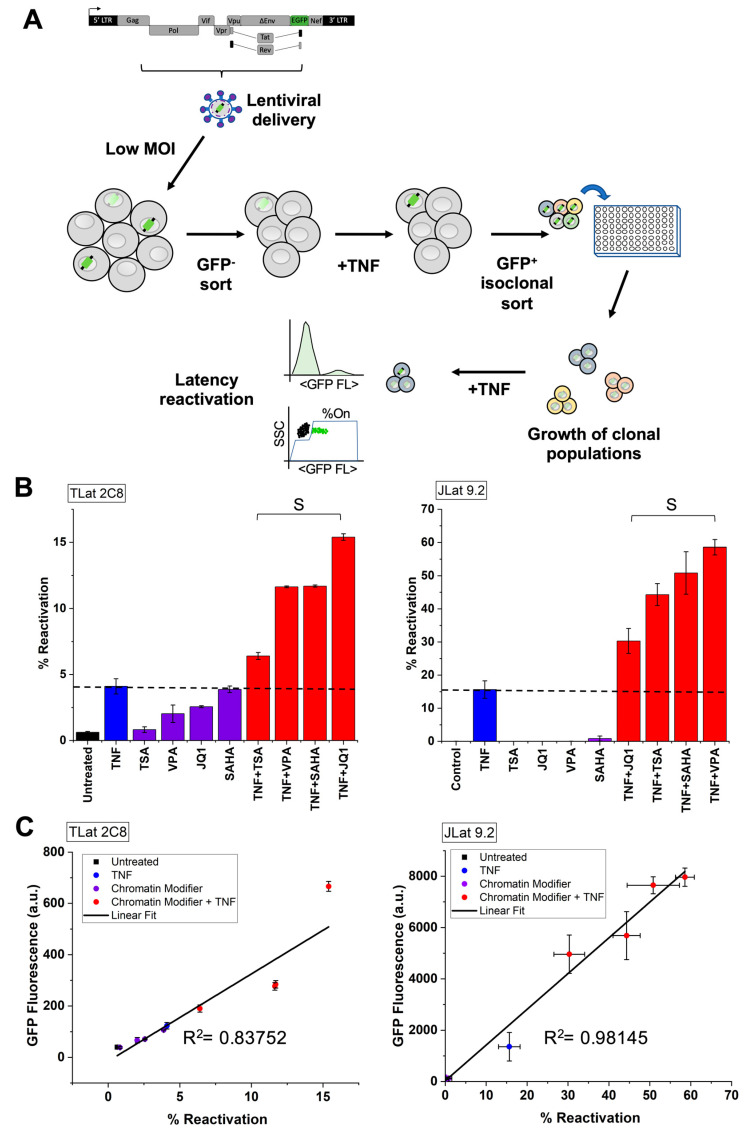
Chromatin modifiers synergize with TNF to enhance reactivation from latency in an HIV latently-infected monocytic cell line. (**A**) A full-length HIV retroviral vector expressing enhanced GFP (EGFP) (NL4-3 ΔEnv EGFP Reporter Vector) was integrated into THP-1 monocytes via lentiviral-mediated delivery at an MOI = 1 (Methods). GFP- cells were sorted by FACS and stimulated with TNF for 24 h. Following TNF stimulation, GFP+ cells were sorted and allowed to relax back into a latent state to generate a clonal cell line. Clonal cell lines were analyzed for GFP expression under basal and TNF-stimulated conditions. The responsive THP-1 latency cell line was termed the TLat. (**B**) To explore HIV reactivation from latency in monocytes, the TLat was exposed to TNF, chromatin modifiers (TSA, VPA, JQ1, and SAHA), and combination treatments for 24h. All drug treatments stimulate HIV reactivation from latency. Synergy is calculated relative to TNF-only using EoB. “S” denotes synergy. All chromatin modifiers synergize with TNF to reactivate HIV from latency in monocytes. The synergistic combination between TNF and JQ1 provides a ~4-fold increase in latent reactivation when compared to TNF-only (Left). Similar to the TLat, combination treatments between chromatin modifiers and TNF synergize to reactivate HIV from latency in JLat 9.2, a well-established HIV latency model in a Jurkat T cell line (Right). Experiments were performed in duplicate and mean and standard error were plotted. (**C**) The mean GFP of the reactivated latent cells positively correlates with reactivation percentage for all drug treatments in monocytes (left) and T cells (right). This shows that cells that turn on more turn on to higher expression levels as opposed to reaching a maximal expression capacity past a reactivation threshold.

**Figure 4 viruses-13-01097-f004:**
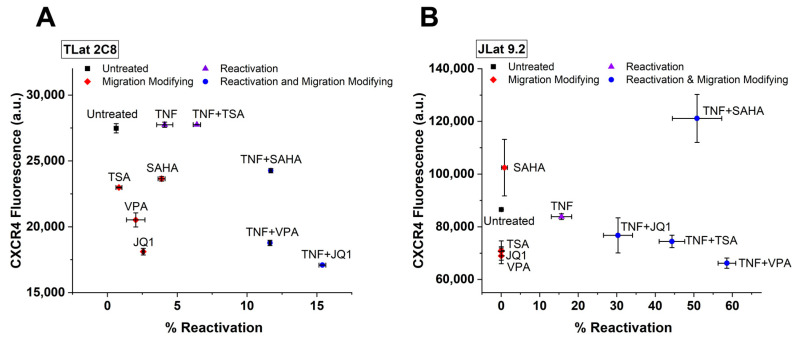
Drug cocktails differentially affect CXCR4 surface expression and reactivation of a latently-infected monocytic cell line. To explore the effects of leading reactivation cocktails on cell migration, CXCR4 surface expression was quantified after 24 h treatment with reactivation cocktails in both (**A**) TLat and (**B**) JLat cell lines. Based on changes in CXCR4 surface expression, the LRA cocktails were classified under three categories: reactivation, migration modifying, and reactivation and migration modifying cocktails (legend). Experiments were performed in duplicate and mean and standard error were plotted.

## Data Availability

The datasets generated and/or analyzed during the current study are available from the corresponding author on reasonable request.

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
