# Peer review of "Synergistic Chromatin-Modifying Treatments Reactivate Latent HIV and Decrease Migration of Multiple Host-Cell Types"

_viruses, 2021, doi:10.3390/v13061097_

Round 1
Reviewer 1 Report
See attached

Reviewer 2 Report
The manuscript by Blanco et al. describes the generation of two new types of reporter cell lines to investigate HIV latency. The monocytic cell line THP1 is used as a model system to investigate HIV latency in monocytes using mini-LTR-GFP reporters and a delta-env HIV reporter. The Jurkat T cell line or the HIV latent variant (J-Lat) is used to compare the findings from the THP1 cells to a T cell model.
The novelty of this research is the generation of a monocytic model for HIV latency that relies on a functional Tat-TAR-axis, which the previously generated monocytic model for latent HIV did not have. This can be promoted a bit more in the abstract and discussion.
Major concerns:
1) it is not clear to me how the transcriptional noise (of the HIV LTR promotor) is calculated/measured (indicated as CV2) and this should be explained. However, it is possible that I have misunderstood this and that promotor activity is purely based on a computational simulation. If this I the case, then this should be clearly stated throughout the manuscript and abstract because now it reads as if the information comes from a measurement using the generated cells lines, but I can’t find out how this is done.
2) the migration studies are not performed using the THP1 cells. Instead, the authors measure CXCR4 expression by flow cytometry on THP1 cells and on Jurkat T cells and then for the Jurkat T cells, a migration assay is done. From the Jurkat experiments, the authors conclude that CXCR4 expression levels are directly proportional to migration (lines 440-442 and figure S5). Unfortunately, figure S5 does not support this claim and this would likely be visible if the CXCR4 fluorescent intensity was plotted on the Y-axis with the migration of Jurkat cells on the X-axis, instead of plotting the information in two separate graphs (the %GFP+ Jurkat cells is irrelevant in this context). By eye-balling the CXCR4 fluorescent intensity on the Jurkats after LRAs, the order from high to low is: 1/2: control/TNF, 3: TNF+JQ1, 4: VPA, 5: JQ1, 6: TNF+VPA, 7: PMA. Then the migration from high to low: 1/2: JQ1/TNF, 3: control, 4 VPA, 5: TNF+JQ1, 6: TNF+VPA, 7: PMA. If migration were directly proportional to CXCR4 expression, then the order would be the same in both lists. However, even if CXCR4 expression was directly proportional in the Jurkat T cells, that would not mean that this would be the same for monocytes as these cells could behave differently and are a cell line used as a mimic for blood circulating monocytes. Thus, the migration exp needs to be performed using the THP1 cells to make the claim that the treatment affects monocyte migration.
Other points:
- I recommend replacing “host-cell types” to “cell types” and “host-cell” to “cell” throughout the manuscript.
- Introduction first few lines; explain that HIV integrates in the genome.
- A promotor is active or activated and a gene is expressed or over-expressed. Alternatively, there can be expression from a promotor but a promotor cannot be over-expressed. This needs to be rephrased throughout the manuscript.
- Using a moi of 1 is understandable in this context but the statement that this is a low moi could be debated among virologists. I recommend omitting the statement that a moi of 1 is a low amount throughout the manuscript.
- Were any of the generated clones confirmed by PCR (either alu or total HIV-DNA)?
- Line130-133: what is the other plasmid that is used to transfect the delta-env HIV construct? The text states that the HEK293 cells are co-transfected.
- All flow experiments and flow methodology: a viability stain needs to be used to select for viable/live cells. If this was not included in the experiment, then the text should be rephrased as such to explain that forward and side scatter were used to select for total cells.
- 2.8 Extrinsic noise filtering and auto-fluorescence correction. It is not clear to me when this correction is applied. Is this to all flow measurements, including the reactivation? Or is this used to determine the CV2? If so, then this should be specified.
If this is indeed the measurement for promotor bursting, then at what timepoint after treatment (I’m assuming this is LRA treatment) are the cells collected for flow analysis? With a less than 2,5 h half life of the GFP, the time point seems quite important.
- Line 222-223: any protein or GFP?
- Lines 225-256: “we also performed extensive simulations to explore the dependence of reactivation (% cells crossing active replication threshold) on changes in …” replace active replication by GFP detection or %cells becoming GFP+, as this is what is measured when using a replication incompetent (delta-env) virus or LTR-reporter constructs.
- Have the authors considered looking at CCR5 upregulation for cell migration purposes? This may be more interesting than CXCR4 as most HIV variants are CCR5 tropic.
- 3.6. This paragraph reads as if the ‘migration’ cocktails induce migration, but these termed cocktails lower CXCR4 expression and will therefore hypothetically reduce migration. Perhaps naming them ‘anti-migration’ cocktails would be an idea.
- To make a solid conclusion on the effect of the tested LRAs on monocyte migration, the migration experiment needs to be performed.
- Lines 439-440. This sentence states that only SAHA affects monocytes and T cells differently in terms of CXCR4 expression. But when I look at the two graphs, I see that there are more differences (as compared to the control): TNF+TSA decrease CXCR4 expression on the Jurkats but not on the monocytes, and TNF+SAHA increasesCXCR4 on the Jurkats but decrease CXCR4 on the mono’s.
- In the discussion the authors elaborate on the possibility that the reactivated virus may impact CXCR4 expression, via nef or other viral proteins. Is the delta-env virus that is used in this study nef competent? Additionally, considering that only cell lines are used in this study, it should be relatively easy to answer the question whether it is the LRA that affects cell migration or the reactivated virus. This can be done by performing a migration experiment with the wild-type cell line treated with the different LRAs.
Round 2
Reviewer 2 Report
I'd like to thank the authors for their thorough and accurate reply to my questions and comments. Every point I made has been addressed appropriately, it is much appreciated.